# Fear of COVID-19 and Perceived Stress: The Mediating Roles of Neuroticism and Perceived Social Support

**DOI:** 10.3390/healthcare10050812

**Published:** 2022-04-27

**Authors:** Qiuyi Yang, Penkarn Kanjanarat, Tinakon Wongpakaran, Chidchanok Ruengorn, Ratanaporn Awiphan, Surapon Nochaiwong, Nahathai Wongpakaran, Danny Wedding

**Affiliations:** 1Master of Science (Mental Health), Graduate School, Chiang Mai University, Chiang Mai 50200, Thailand; qiuyi_yang@cmu.ac.th (Q.Y.); chidchanok.r@cmu.ac.th (C.R.); ratanaporn.a@cmu.ac.th (R.A.); nahathai.wongpakaran@cmu.ac.th (N.W.); danny.wedding@gmail.com (D.W.); 2Department of Pharmaceutical Care, Faculty of Pharmacy, Chiang Mai University, Suthep Road, T. Suthep, A. Muang, Chiang Mai 50200, Thailand; surapon.nochaiwong@cmu.ac.th; 3Department of Psychiatry, Faculty of Medicine, Chiang Mai University, 110 Intawaroros Rd., T. Sriphum, A. Muang, Chiang Mai 50200, Thailand; 4Department of Clinical and Humanistic Psychology, Saybrook University, Pasadena, CA 91103, USA; 5Department of Psychology, University of Missouri-Saint Louis, St. Louis, MO 63121, USA

**Keywords:** COVID-19, fear of COVID-19, perceived stress, neuroticism, perceived social support, mediation analysis

## Abstract

Background: Fear of COVID-19 leads to stress and may result in various kinds of mental health problems. Many factors are associated with an individual’s perception of stress, including neuroticism and perceived social support. This study aimed to examine the role of neuroticism and perceived social support as mediators of fear of COVID-19 on perceived stress. Methods: Data from 3299 participants aged ≥18 years from the HOME-COVID-19 survey in 2020 were used for analysis. Measurements used included the Fear of COVID-19 and Impact on Quality of Life Scale, the Perceived Stress Scale-10, the Neuroticism inventory and the Multidimensional Scale of Perceived Social Support-12. A parallel mediation model within a structural equation modeling framework with 5000 bootstrapping sampling was used to test the mediating effect. Results: Fear of COVID-19 had a direct effect on perceived stress (B  =  0.100, 95% CI = 0.080–0.121, *p* < 0.001), whereas neuroticism, but not perceived social support, partially mediated the relationship between fear of COVID-19 and perceived stress (B = 0.018, 95% CI = 0.000–0.036). Among all types of social support, only perceived support from friends was a significant mediator (B = 0.016, 95% CI = 0.006–0.025). Conclusions: Neuroticism and perceived support from friends are critical factors in the relationship between fear of COVID-19 and perceived stress.

## 1. Introduction

Since the end of 2019, COVID-19 (coronavirus disease 2019) has been spreading globally. The ongoing COVID-19 pandemic affects the mental health of individuals and society directly and indirectly [1]. A recent report estimated the 2020 worldwide prevalence of stress symptoms, anxiety, stress response, depression, psychological distress and sleep problems at 24.1, 26.9, 36.5, 28.0, 50.0m and 27.6%, respectively [2]. Populations at higher risks of mental health problems include health workers, people with non-communicable chronic diseases, individuals with pre-existing mental health disorders, the elderly, people in isolation, patients with COVID-19, homeless people and refugees [2,3].

COVID-19 pandemic has caused fear in people globally [4,5,6]. Fear of COVID-19 is described as fear of contracting COVID-19 by living with or near a person with COVID-19, failing to practice general prevention of COVID-19 or receiving or giving blood [7]. Excessive fear during crisis situations may produce harmful effects [5,6,8]. At a personal level, excessive fear leads to excessive stress, anxiety, depression and insomnia [4,5,6,9]. At the social level, fear might lead to stigma, discrimination or even hate crimes against specific groups of people, psychological racism and xenophobia [5,6,10]. Individuals with less fear might engage in risky behaviors and increased social activity, such as going out without a face mask or hosting a party [5,6,8,11]. In contrast, fear of COVID-19 can also trigger safe behaviors such as wearing a mask, washing hands and avoiding social events, which mitigate certain threats such as risk of COVID-19 infection [12,13]. 

When fear arises, people experience stress. Perceived stress is an individual response or experience to a stimulus, threat or event in different circumstances, which is one of the gateways to several mental health disorders, such as anxiety, depression, phobia, post- traumatic disorder and even psychotic disorders [14,15,16,17]. Stress affects those who are predisposed to psychological health problems but also the general population (e.g., it can affect sleep, emotional eating and weight gain) [18,19,20]. The COVID-19 pandemic has brought forth higher degrees of perceived stress globally [2,21,22,23,24]. The prevalence of perceived stress was higher among health workers than that of the general population (44.86% and 29.6%, respectively) [25,26]. Fear of infection, fear of transmission to family members and several other factors were the main stressors, and better self-rated health was one of the main predictors of lower perceived stress [13,27,28,29,30]. In addition, a higher level of perceived stress and higher fear of COVID-19 produced lower levels of happiness, life satisfaction and quality of sleep [31,32,33,34].

There are certain personality traits that are associated with perceived stress and fear. Neuroticism, the tendency of a relatively stable negative emotional reaction to threats, frustration or loss, is one of them [35,36]. Neuroticism is a significant public health problem [35,36], which is closely related to different forms of psychopathology, and it predicts substance abuse, depression and anxiety disorders [35,36,37]. During the COVID-19 pandemic, neuroticism was reported to be a significantly correlated personality trait that predicted fear of COVID-19 among 7309 participants from 96 countries [38]. A recent study in Eastern Europe found that neuroticism increased as a result of over focus on the pandemic and educational burnout, particularly in young people [39]. Many studies have linked neuroticism to higher levels of stress [38,40,41,42,43]. Individuals with higher neuroticism levels always have greater health risks at the beginning of a pandemic [43,44].

Social support is one of the important components of health, described as individuals caring for, respecting and supporting each other as a part of a social network. Social support is associated with lower morbidity and mortality. Social support reduces psychological distress, adjusts chronic stress states and promotes psychological adjustment [45,46]. A significant negative correlation between perceived social support and fear of COVID-19 in college students was documented in two studies [47,48], while two others reported the opposite finding [49,50]. Perceived social support was also negatively correlated with perceived stress during the COVID-19 pandemic [51]. In contrast, one study found a weak positive correlation between the two [49,51]. 

Little is known about how neuroticism and perceived social support serve as mediators influencing the association between fear of COVID-19 and perceived stress. No previous investigators have studied the association with perceived stress and fear of COVID-19 using perceived social support as a mediator. The present study aimed to examine the roles of perceived social support and neuroticism in the mediation of the relationship between perceived stress and fear of COVID-19. We hypothesized that there were both direct and indirect effects of the fear of COVID-19 concerning perceived stress through the two mediators in the general Thai population.

## 2. Materials and Methods

The present study employed a secondary data analysis of The Health Outcomes and Mental Health Care Evaluation Survey: Under the Pandemic Situation of COVID-19 (HOME-COVID-19), which was a cross-sectional survey with Wave I data collected from 21 April to 4 May 2020, from 4004 participants comprising the general population in Thailand [7]. Participants aged 18–59 years old were included in this analysis.

The HOME-COVID-19 study.

This was an open, voluntary and nationwide cross-sectional online survey using the SurveyMonkey^®^ platform in Thailand. Participants included Thai citizens, permanent residents and non-residents with employment or work permits, all aged 18 years or above at the time of the survey. The study recruited participants using convenience sampling and a snowball technique by posting the survey QR codes or links to social media networks and public websites, such as Facebook, LINE, Twitter and Instagram. The study was in line with the Strengthening the Reporting of Observational Studies in Epidemiology Statement [52] and Improving the Quality of Web Surveys: The Checklist for Reporting Results of Internet E-Surveys [53]. Details of the methods of HOME-COVID-19 have been published elsewhere [7].

Because the present study comprised a secondary data analysis, the authors were approved for ethics exemption by the Ethics Committee of the Faculty of Pharmacy, Chiang Mai University (Deliberation number: 001/2022/Exe).

### 2.1. Measurements

Sociodemographic data were collected by self-report—i.e., age, sex, marital status, education level, income, religion, regions of residence, occupation and work status, health status and living status (e.g., being in quarantine).

#### 2.1.1. Fear of COVID-19 and Impacts on Quality of Life Scale

The scale comprises two parts, namely fear of COVID-19 (9 items) and impacts on quality of life (QoL) (8 items). Regarding the fear of COVID-19 (FOC) part, each item on a five-point Likert-type scale, ranging from 4 (I am the most fearful) to 0 (I am not fearful at all) and indicated the level of fear for each condition. A higher score indicates a higher FOC. For the Impact on QoL part, each item on a five-point Likert-type scale, ranging from 4 (Most impacted) to 0 (Least or not at all impacted), indicated the level of impact of the fear. A higher score indicates a higher impact of fear. The total scores range from 0 to 68 with 0–36 in the FOC part and 0–32 in the impacts on the QoL part. A pilot study among 30 adult Thai participants demonstrated Cronbach’s alphas of 0.925 and 0.911 in the FOC parts and impacts on QoL, respectively [54]. In the present study, the Cronbach’s alpha of FOC was 0.878. 

#### 2.1.2. Perceived Stress Scale-10 (PSS-10)

The 10-item scale was developed to measure perceived stress [55]. It contains two factors—i.e., 6 items for stress and 4 items for control. Each item is scored on a five-point Likert-type scale, ranging from 4 (very often) to 0 (never) and indicated the frequency of feeling or thinking a particular way during the past month. A higher score indicates higher perceived stress, and scores range from 0 to 40. The Thai version was validated with a Cronbach’s alpha of 0.85, excellent goodness-of-fit and good validity [21]. Cronbach’s alpha for the present study was 0.821.

#### 2.1.3. Neuroticism Inventory (NI)

The Neuroticism Inventory (NI) was developed by Wongpakaran et al. and comprises 15 items, with a Cronbach’s α of 0.83 [16]. Each item is scored on a four-point Likert-type scale, ranging from 4 (Always like me) to 1 (Never like me). Scores range from 15 to 60, with higher scores indicating higher neuroticism. Cronbach’s alpha for the present study was 0.917.

#### 2.1.4. Multidimensional Scale of Perceived Social Support (MSPSS)

The MSPSS has been widely used globally [56]. The psychometric properties and factor structure of the revised Thai version were developed and studied in adult Thai samples [54]. Each item is scored on a seven-point Likert-type scale, ranging from 7 (very strongly agree) to 1 (very strongly disagree) and indicates the perception of how much exterior social support was provided by an individual. The scale demonstrates three sources of support (subscales): family (items 3, 4, 8 and 11), friends (items 6, 7, 9 and 12) and significant others (items 1, 2, 5 and 10). A higher score indicates higher perceived social support, with total scores ranging from 12 to 84, and each subscale ranging from 4 to 28. Cronbach’s alpha was 0.87. The present study reported Cronbach’s alphas of 0.913, 0.903, 0.916, and 0.888 for total, support from family, friends, and significant others, respectively.

### 2.2. Data Cleaning

A total of 4004 eligible participants completed the online survey, and a related study displayed details of study participants [57]. Data were checked for multivariate normality, with 89 cases with standardized residuals greater than ±2.5 excluded. Then, Mahalanobis distance was used to detect the outliers at *p*-value < 0.001, and 432 cases were excluded. After data exploration, we found that there were small proportions of participants with unidentified gender and aged 60 years or older. Thus, we decided to further exclude 136 participants with unidentified gender and 49 participants aged 60 or older. Finally, 3299 participants were included in the analysis.

### 2.3. Data Analysis

Characteristics of the participants were analyzed using descriptive statistics. Perceived stress, fear of COVID-19, neuroticism and perceived social support are presented as mean, standard deviation and range (min–max). The covariates in the present study, including age, sex, marital status, education level, living status, religion, region of residence, occupation status, work from home, income, history of chronic non-communicable disease, health insurance and quarantine status were analyzed by number and percentage. 

Pearson’s correlations were analyzed to identify associations between each pair of psychological variables, i.e., fear of COVID-19, perceived stress, neuroticism, total scale of perceived social support and the subscales of perceived social support from significant others, family members and friends. To identify associations between psychological variables and potential covariates to include in mediation analyses, we performed parallel multiple mediation analysis. Structural equation modeling was performed for mediation analyses to identify direct effect of fear of COVID-19 on perceived stress and indirect effect of fear of COVID-19 on perceived stress via neuroticism and perceived social support. To examine model fitness, the following fitness indices were used: the comparative-fit index (CFI), >0.95; the Tucker–Lewis Index (TLI), >0.95; the root mean square error of approximation (RMSEA), <0.06 [58]. The model was tested using a maximum-likelihood estimation method for covariance matrices. Unstandardized regression coefficients and *p*-values were reported for the coefficients and bootstrap confidence intervals for the conditional indirect effects. Confidence intervals that did not include zero were indicative of statistical significance. 

Hypothesized model.

We hypothesized that there were positive correlations between fear of COVID-19, perceived stress and neuroticism, while there were negative correlations between fear of COVID-19 and perceived social support, between perceived social support and perceived stress and between perceived social support and neuroticism. We also hypothesized that there was a significant direct effect of fear of COVID-19 and perceived stress and a significant indirect effect of fear of COVID-19 and perceived stress through neuroticism and perceived social support (parallel mediation) (Figure 1).

All significant covariates were controlled in the mediation model. We performed mediation analyses of overall perceived social support using the total scale (Model 1) and subscales from each source of perceived social support, i.e., support from significant others (Model 2), family members (Model 3) and friends (Model 4). 

Stata 14 statistical software was used for all analyses in the present study, with significance levels set at <0.05. 

## 3. Results

### 3.1. Sociodemographic and Psychological Characteristics of Participants

As shown in Table 1, of 3299 participants (mean age, 28.52  ±  9.9 years), 1025 (31.1%) were male and 2274 (68.9%) were female, 1679 (50.6%) participants had a bachelor’s degree or higher education level, 2715 (82.3%) participants were single and 2864 (86.8%) of the participants were Buddhist. The average perceived stress, fear of COVID-19, neuroticism and perceived social support scores were 17.61 ± 5.76, 20.84 ± 7.07, 36.27 ± 9.57 and 59.01 ± 13.48, respectively.

### 3.2. Psychological Variables and Characteristics of Participants

As shown in Table 2, age, sex, education level, religion, marital status, region of residence, income and quarantine status were significantly associated with perceived stress, neuroticism and fear of COVID-19 at *p* < 0.05. In contrast, occupation, education level, religion, health insurance, work from home and history of chronic non-communicable disease were related to perceived social support.

### 3.3. Pearson’s Correlation among Psychological Variables and Multiple Regression Analyses

In Table 3, fear of COVID-19 negatively correlated with perceived social support (r = −0.037, *p* < 0.01) and positively with perceived stress (r = −0.176, *p* < 0.001) and neuroticism (r = 0.050, *p* < 0.001). Notably, perceived stress had a stronger positive correlation with neuroticism (r = 0.685, *p* < 0.001) and a negative correlation with perceived social support (r = −0.381, *p* < 0.001). Moreover, neuroticism negatively correlated with perceived social support (r = −0.330, *p* < 0.001). The subscales with three different sources of perceived social support had similar results to the total scale with other psychological variables.

From multivariable regression analysis, fear of COVID-19, perceived social support, neuroticism and the following covariates, i.e., age, sex, education level, religion, occupation status, history of chronic non-communicable disease and quarantine status were shown to be associated with perceived stress (R^2^ =0.533, F (16, 3282) = 234.42, *p* < 0.001). These variables were included in the mediation analysis. 

### 3.4. Mediation Analysis

The direct, indirect and total effects of fear of COVID-19 on perceived stress, while considering neuroticism and perceived social support as a presumed mediator, are displayed in Table 4. The mediation analysis is visually presented in Figure 2, Figure 3, Figure 4 and Figure 5. First, as reported in Figure 2 (Model 1), when controlling covariates, neuroticism, the first mediator, was regressed on perceived stress and fear of COVID-19 (paths a1,b1). Results showed a significant positive association between neuroticism and fear of COVID-19 (B = 0.048, *p* = 0.047) as well as perceived stress (B = 0.347, *p* < 0.001). The second mediator, perceived social support, was regressed on fear of COVID-19 (path a2), which was not significant, *p* > 0.05. When perceived social support was regressed on perceived stress (B = −0.074, *p* = 0.001), a significantly negative path (b2) was revealed. Then, we analyzed the subscale of perceived social support, i.e., social support from significant others, family members and friends in the parallel mediation model (Model 2/Model 3). The results indicated that fear of COVID-19 was not significantly associated with perceived social support from family members and significant others when controlling covariates, *p* > 0.05. Fear of COVID-19 was only significantly associated with perceived social support from friends (B = −0.047, *p* = 0.001). Through two mediators, neuroticism and perceived social support from friends, we observed a strong positive effect both for neuroticism (a1b2 = 0.018, 95%CI = 0.000–0.036) and for perceived social support from friends (a2b2= 0.016, 95%CI = 0.006–0.025). In Figure 5 (Model 4), the direct effect of fear of COVID-19 on perceived stress (c’) was 0.100 (95%CI = 0.080–0.121, *p* < 0.001). The total indirect effect through two mediators (a1b1 + a2b2) was 0.023 (95%CI = 0.005–0.042, *p* = 0.013).

Regarding the model fitness, all models were shown to have the best fit model, resulting in the following fitness statistics: CFI = 1.000, TFI = 1.001, RMSEA < 0.001. 

In summary, neuroticism and perceived social support from friends were significant mediators of the relationships between fear of COVID-19 and perceived stress when controlling covariates.

## 4. Discussion

The major objective of the study was to test whether and how fear of COVID-19 affects perceived stress during the primary stage of the pandemic in Thailand. Consequently, neuroticism and perceived social support were studied as presumed mediators in that relationship. The present study documented a parallel model of neuroticism and perceived social support from friends that mediated the relationship between fear of COVID-19 and perceived stress. 

In line with other related studies, the results of this research indicated that fear of COVID-19 positively correlated with perceived stress and neuroticism [31,32,33,38,49,59,60,61,62,63,64]. These results confirmed the findings from other countries during the pandemic. Fear of COVID-19 directly affected perceived stress in the present study. This suggests that aggravated fear of COVID-19 was related to increased perceived stress levels in the Thai population. Infectious diseases can cause instinctive fear [5,6,65,66], and fear of COVID-19 varies in different regions and different populations [5,33,34,49,50,62,67,68,69,70,71]. At the same time, heightened fear is concerned with the social learning of fear, in that fear can be learned through other individuals or social media [72], and people repeatedly exposed to COVID-19 information express more fear of COVID-19 [57]. Another significant factor involved the associated state quarantine, which may have exacerbated their fear [5,8,72,73]. A partial intersection was observed between stress and fear mechanisms [74]. The increasing fear of COVID-19 level activates those parts of the brain that trigger the release of stress hormones such as cortisol. These, together with the loss of economic resources and uncertainty about the future, put considerable pressure on the population studied [74,75].

In line with previous studies [47,48,76,77], individuals with higher perceived social support had lower perceived stress. Perceived social support serves as a significant factor in psychological adjustment, leading to reduced stress [78,79]. Findings from other studies among Chinese students have found that social support can prevent negative psychological and emotional effects [80,81] Perceived social support from friends was the most negatively correlated with fear of COVID-19. This could be because most of the participants in the study were young. Previous evidence demonstrated that the mere perceived existence of social support, or even just reminders of such, could decrease physical and psychological responses to threats [82]. The mechanism of this effect remains unclear, but we speculate that social support may reduce individuals’ physical and psychological responses to COVID-19 threats, consequently reducing fear of COVID-19 [47,48,51,76,77]. 

We found a negative correlation between neuroticism and perceived social support, which was in line with previous research [83,84]. Higher neurotic individuals may experience lower perceived social support because their poor relationship satisfaction leads to limited socialization due to their characteristic negative emotions [85,86].

Neuroticism mediated the relationship between fear of COVID-19 and perceived stress in the present study. In general, individuals with higher neuroticism levels tend to have higher perceived stress levels when exposed to the same levels of fear of COVID-19 in Thailand. Neuroticism was not only positively associated with fear of COVID-19 but was positively associated with panic buying and fear of death during the pandemic [87,88]. The findings from the present study reinforce the previous finding that neuroticism was associated with stronger emotional responses, poorer emotional perceptions, emotional coping and emotional experience [89]. Neuroticism was also found to be the most significant factor influencing the perception of threat from COVID-19 [38]. Further, the relationship between neuroticism and stress was the same as that in studies reported during the pandemic [40,42,89]. Individuals with higher neuroticism levels also focused more on COVID-19-related information and worried more about the aftermath of the pandemic [41]. These factors all led individuals to experience higher levels of neuroticism, more negative emotions and higher stress levels during the pandemic [41,42,90]. Some longitudinal studies have found that neuroticism increases the possibility of stressful experiences in the lifestyles of highly neurotic individuals, and these stressors in turn trigger common mental disorders [91,92,93]. Longitudinal data also showed that higher neuroticism levels resulted in higher levels of subjective stress and cortisol, and that neuroticism could predict the accumulation of biological stress during the COVID-19 period [43].

The current study is the first to discover that perceived social support from friends has a mediating role between fear of COVID-19 and perceived stress. Family and friends provide practical and emotional support when individuals need help coping with stressful life events [94]. Particularly during the COVID-19 pandemic, the government implemented social distancing policies (e.g., work from home, online study, or even lockdown) [28,73,95]. Subsequently, people spent more time with their families at home or with friends who live together. For young adults, social support from a friend had an important impact on the psychological effects of the pandemic. Previous studies of Chinese students found that peer support was an important source of social support for college students [80,81]. We speculate that perceived social support from family did not moderate the relationship between fear of COVID-19 and perceived stress, but rather had a direct effect on perceived stress due to high perceived social support in Thai adults in the present study, even during the pandemic. A study of Thai adults during the early phase of the COVID-19 pandemic suggested that about 40–58% of the participants reported better relationships with family by providing care, communicating, offering emotional support, supporting problem-solving for each other and participating in family activities [96]. Direct association of perceived social support from family on perceived stress during the pandemic in adult populations has been reported in adult populations in different countries [77,97]. In addition, numerous studies have shown that perceived social support plays a role as a moderator or regulator of stress [82,98,99] by altering the hypothalamic–pituitary–adrenal axis, which is associated with the stress response system [100,101]. A moderation effect of perceived social support was not tested in the present research, but we will investigate this moderation effect in a future analysis. 

Compared with participants from other countries, the Thai participants exhibited lower perceived stress and fear of COVID-19 at the beginning of the pandemic [13,34,68,95,102,103]. Among Southeast Asian countries, the first COVID-19 cases outside of China were reported in Thailand [104]. During the data collection period, Thailand was at the primary stage of the COVID-19 pandemic, with about 3000 confirmed cases reported, of which most were concentrated in Bangkok [105]. The situation was not considered as serious as in other countries during this period [106]. The Thai government implemented lockdown measures nationwide on April 3, 2020 (e.g., declaring a nationwide curfew, requiring people to wear masks, maintaining social distancing and enforcing nighttime curfew in some areas), which effectively stopped the viral transmission and the number of new cases continued to decline [104]. These may be the reasons that Thais were less fearful of COVID-19 than those in other countries during this period. Regarding other sociodemographic variables, it is anticipated that fear, perceived stress and neuroticism, but not perceived social support, are associated with advanced age, female gender and low income. However, for other variables, i.e., religion, occupation and work from home, the findings cannot be simply explained. 

### 4.1. Strengths and Limitations

The present study is the first investigation of the relationship between COVID-19 fear and perceived stress though neuroticism and perceived social support. The sample size is large. 

The study has some limitations. First, the current study design is cross-sectional, and it cannot offer definitive conclusions about causal effects among fear of COVID-19, neuroticism, perceived social support and perceived stress. Second, the online and self-reported surveys may contain response biases. In particular, the study design did not allow us to reach residents without Internet access who might have suffered during the pandemic. Not only that, but some participants with higher perceived stress might not have been able to complete the questionnaire. Finally, the study did not address whether families or friends tested positive for COVID-19; this would obviously affect fear of COVID-19 and perceived stress. 

### 4.2. Practical Implications

In order to reduce perceived stress as a result of fear of COVID-19 during the beginning of the pandemic, governments must provide sufficient and accurate information about the transmission and preventive actions to the public to counterbalance false information that triggers more fear. In addition, all sectors of society should carry out education campaigns to further disseminate public health information and prevent the spread of the disease; in particular, signs of perceived stress should be communicated to the public. If this was done, people could easily and earlier identify the perceived stress experienced by themselves and their family, friends and social network. In addition, healthcare providers, family and friends should observe and monitor stress from fear of COVID-19, particularly in individuals at high risk of stress, such as individuals with neuroticism personality traits, and prioritize support to these individuals. Although, there was some limitation to social gathering during the pandemic, for young adults it is important to keep connections and communication with close friends and family members through available methods (e.g., social media) to provide mutual support. Political leaders should organize alternative quarantine protocols for people who experienced fear and stress and should arrange for them to stay with family or friends. Moreover, cooperation between family, friends and significant others, including schoolteachers, counselors, colleagues, healthcare providers and community leaders to provide instrumental, emotional support and reliable information support to reduce stress during the pandemic are crucial to prevent future mental health problems.

## 5. Conclusions

These findings suggest that neuroticism, fear of COVID-19 and perceived social support affected perceived stress in adults during the early stage of the COVID-19 pandemic. The most important finding was that social support from friends proved particularly important for young people, and highly neurotic individuals were at higher risk for stress during the COVID-19 pandemic. In addition, neuroticism and perceived social support from friends significantly mediated the relationship between fear of COVID-19 and perceived stress. Specifically, higher levels of neuroticism were related to higher levels of fear of COVID-19 and perceived stress. Higher levels of perceived social support from friends were related to lower levels of fear of COVID-19 and perceived stress. In the future, an intervention related to providing support from friends might be designed to decrease the perception of stress, especially among young people during the COVID pandemic. 

## Figures and Tables

**Figure 1 healthcare-10-00812-f001:**
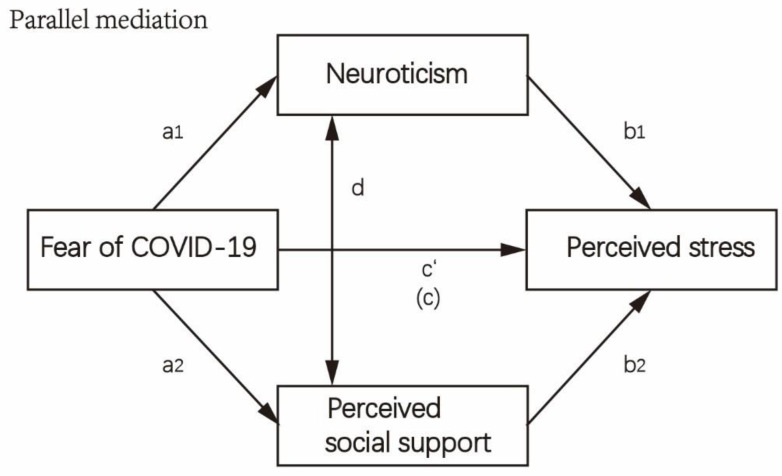
Hypothetical parallel mediation model description. The coefficient “a1” symbolizes the regression coefficient between fear of COVID-19 and neuroticism; the coefficient “b1”, between neuroticism and perceived stress. The coefficient “a2” refers to the sectional regression coefficient between fear of COVID-19 and perceived social support; the coefficient “b2”, between perceived social support and perceived stress; and the correlation “d”, between neuroticism and perceived social support. The coefficient “c’” represents the relationship between fear of COVID-19 and perceived stress when the mediators are added. The coefficient “c” stands for the coefficient in the regression model without the mediators.

**Figure 2 healthcare-10-00812-f002:**
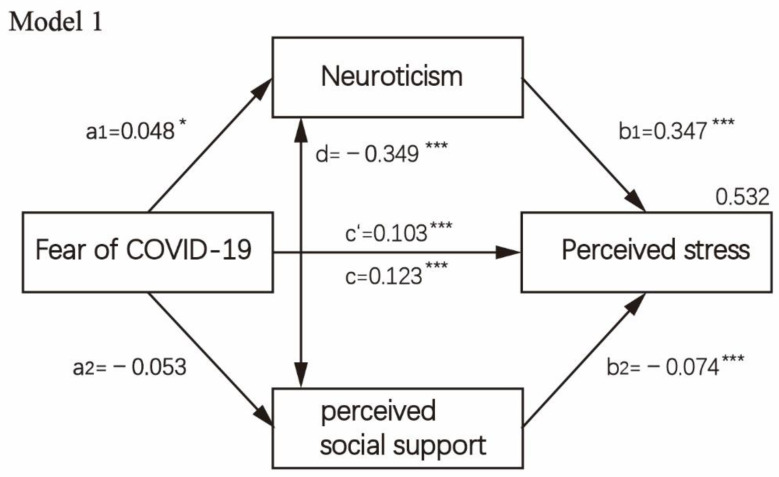
Mediation analysis test on fear of COVID-19 concerning perceived stress and mediating role of neuroticism and perceived social support and controlling covariates, i.e., age, sex, education level, religion, occupation status, history of chronic non-communicable disease and quarantine status (*n* = 3299). Note: * *p* < 0.05; *** *p* < 0.001.

**Figure 3 healthcare-10-00812-f003:**
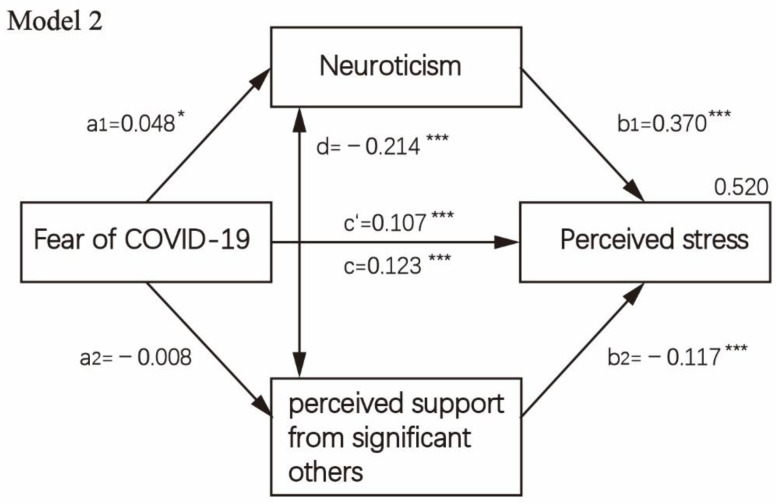
Mediation analysis test on fear of COVID-19 concerning perceived stress and mediating role of neuroticism and subscale of perceived social support from significant others and controlling covariates, i.e., age, sex, education level, religion, occupation status, history of chronic non-communicable disease and quarantine status (*n* = 3299). Note: * *p* < 0.05; *** *p* < 0.001.

**Figure 4 healthcare-10-00812-f004:**
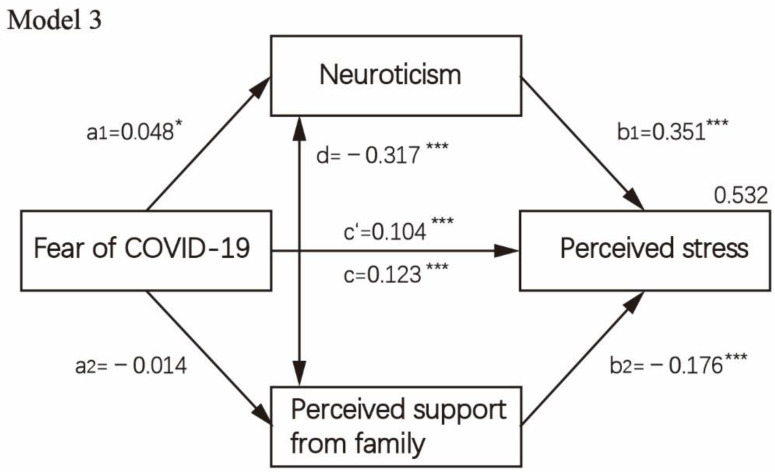
Mediation analysis test on fear of COVID-19 concerning perceived stress and mediating role of neuroticism and subscale of perceived social support from family members and controlling covariates, i.e., age, sex, education level, religion, occupation status, history of chronic non-communicable disease and quarantine status (*n* = 3299). Note: * *p* < 0.05; *** *p* < 0.001.

**Figure 5 healthcare-10-00812-f005:**
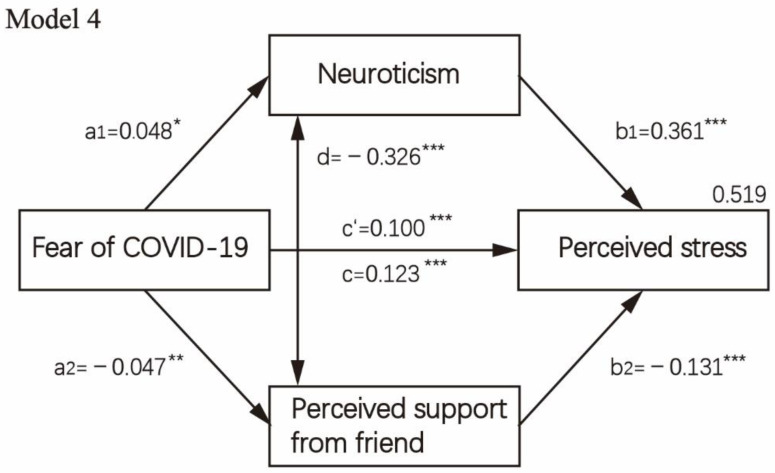
Mediation analysis test on fear of COVID-19 concerning perceived stress and mediating role of neuroticism and subscale of perceived social support from friends and controlling covariates, i.e., age, sex, education level, religion, occupation status, history of chronic non-communicable disease and quarantine status (*n* = 3299). Note: * *p* < 0.05; ** *p* < 0.01; ***, *p* < 0.001.

**Table 1 healthcare-10-00812-t001:** Participant characteristics by percentage, mean and SD (*n* = 3299).

Variables	Mean (SD) or *n* (%)
Age (Years), 18–59 Mean (SD)	28.52 (9.88)
Sex, *n* (%)	
Male	1025 (31.1)
Female	2274 (68.9)
Education level, *n* (%)	
Bachelor’s degree or higher	1670 (50.6)
Below bachelor’s degree	1629 (49.4)
Marital status, *n* (%)	
Single	2715 (82.3)
Non-single	584 (17.7)
Religion, *n* (%)	
Buddhist	2864 (86.8)
Non-Buddhist	435 (13.2)
Occupation, *n* (%)	
College student	1313 (39.8)
Self-employed/private-enterprise employee	776 (23.5)
Farmer/laborer freelance/others	467 (14.1)
Government/state-enterprise employee	430 (13.1)
Unemployed/retired	313 (9.5)
Region of residence, *n* (%)	
Capital and vicinity	1153 (35.0)
Northern	1083 (32.8)
Central	396 (12.1)
Southern	224 (6.8)
Northeastern	222 (6.7)
Eastern	143 (4.3)
Western	78 (2.3)
Living status, *n* (%)	
Live with family	2599 (78.8)
Live alone	474 (14.3)
Live with others	226 (6.9)
Income (Thai Baht), *n* (%)	
≤10,000	1593 (48.3)
>10,000	1706 (51.7)
Health insurance, *n* (%)	
Non-Universal Coverage Scheme	2217 (67.2)
Universal Coverage Scheme	1082 (32.8)
History of chronic non-communicable disease, *n* (%)	
No	2847 (86.3)
Yes	452 (13.7)
Work from home, *n* (%)	
No	733 (22.2)
Yes	2566 (78.8)
Quarantine status, *n* (%)	
Never	1440 (43.6)
Was quarantined	1329 (40.3)
During quarantine	530 (16.1)
Scores of psychological measures, Mean (SD)	
Perceived stress (range 0–36)	17.61 (5.76)
Fear of COVD-19 (range 0–36)	20.84 (7.07)
Neuroticism (range 15–60)	36.27 (9.57)
Perceived social support—Total score (range 12–84)	59.01 (13.48)
-Perceived social support from significant others (range 4–28)	19.09 (5.74)
-Perceived social support from family members (range 4–28)	20.07 (5.60)
-Perceived social support from friends (range 4–28)	19.85 (5.20)

Non-single: married/domestic partnership. Single: single/divorced/widowed/separated. Non-Buddhist: Non-religious/Christian/Muslim/other. Non-Universal Coverage Scheme: state enterprises/social security scheme/self-payment/other.

**Table 2 healthcare-10-00812-t002:** Mean and SD of psychological variables in participant characteristics (N = 3299).

Variables	*n*	Perceived Stress		Fear of COVID-19		Neuroticism		Perceived Social Support	
Age		Mean ± SD	*p*-Value	Mean ± SD	*p*-Value	Mean ± SD	*p*-Value	Mean ± SD	*p*-Value
<21	644 (19.6)	19.10 ± 5.64	<0.001	21.33 ± 6.82	0.017	39.47 ± 8.57	<0.001	58.66 ± 13.41	0.074
21–30	1574 (47.7)	18.31 ± 5.61		20.44 ± 6.95		37.84 ± 9.30		59.50 ± 13.44	
31–40	640 (19.4)	16.61 ± 5.25		21.32 ± 7.22		32.89 ± 9.27		57.79 ± 13.95	
41–50	288 (8.7)	15.27 ± 5.70		21.12 ± 7.61		32.02 ± 9.09		59.37 ± 13.22	
51–60	153 (4.6)	12.77 ± 5.40		20.41 ± 7.58		28.86 ± 7.68		59.86 ± 12.30	
Sex									
Male	1025 (31.1)	15.84 ± 5.55	<0.001	19.44 ± 7.12	<0.001	33.27 ± 9.28	<0.001	58.36 ± 13.17	0.066
Female	2274 (68.9)	18.41 ± 5.68		21.47 ± 6.96		37.63 ± 9.39		59.30 ± 13.60	
Education level									
Bachelor’s degree or higher	1670 (50.6)	18.27 ± 5.66		21.27 ± 6.79		37.43 ± 9.50		58.20 ± 13.49	
Below bachelor’s degree	1629 (49.4)	16.94 ± 5.78	<0.001	20.40 ± 7.33	<0.001	35.09 ± 9.50	<0.001	59.84 ± 13.42	<0.001
Marital status									
Single	2715 (82.3)	17.99 ± 5.72	<0.001	20.56 ± 6.98	<0.001	37.19 ± 9.40	<0.001	59.03 ± 13.32	0.856
Non-single	584 (17.7)	15.88 ± 5.64		22.16 ± 7.37		32.02 ± 9.19		58.92 ± 14.19	
Religion									
Buddhist	2864 (86.8)	17.35 ± 5.69	<0.001	21.05 ± 6.97	<0.001	35.83 ± 9.48	<0.001	59.66 ± 13.13	<0.001
Non-Buddhist	435 (13.2)	19.38 ± 5.89		19.49 ± 7.62		39.18 ± 9.64		54.73 ± 14.89	
Occupation									
College student	1313 (39.8)	18.40 ± 5.73	<0.001	20.55 ± 6.70	0.054	38.51 ± 8.89	<0.001	59.60 ± 12.96	0.040
Other occupations	1986 (60.2)	17.09 ± 5.72		21.03 ± 7.31		34.79 ± 9.72		58.62 ± 13.80	
Region of residence									
Non-capital city and its vicinity	2146 (65.0)	17.37 ± 5.41	0.001	21.44 ± 7.09	<0.001	35.57 ± 9.43	<0.001	59.21 ± 13.25	0.238
Capital city and its vicinity	1153 (35.0)	18.07 ± 6.33		19.72 ± 6.91		37.57 ± 9.71		58.63 ± 13.88	
Residence									
Not residing with family	700 (21.2)	17.68 ± 5.96	0.736	20.12 ± 7.29	0.002	36.52 ± 9.43	0.447	58.13 ± 13.34	0.052
Residing with family	2599 (78.8)	17.60 ± 5.70		21.04 ± 7.00		36.21 ± 9.61		59.24 ± 13.51	
Income (THB)									
≤10,000	1593 (48.3)	18.53 ± 5.67	<0.001	21.34 ± 6.85	<0.001	38.30 ± 9.28	<0.001	58.67 ± 13.42	0.162
>10,000	1706 (51.7)	16.76 ± 5.71		20.37 ± 7.25		34.37 ± 9.45		59.33 ± 13.52	
Health insurance									
Non-Universal Coverage Scheme	2217 (67.2)	17.36 ± 5.79	<0.001	20.73 ± 7.19	0.206	35.83 ± 9.61	<0.001	59.49 ± 13.56	0.003
Universal Coverage Scheme	1082 (32.8)	18.14 ± 5.67		21.06 ± 6.83		37.17 ± 9.43		58.02 ± 13.25	
History of chronic noncommunicable disease									
No	2847 (86.3)	17.54 ± 5.69	0.056	20.72 ± 7.07	0.013	36.21 ± 9.44	0.337	59.50 ± 13.35	<0.001
Yes	452 (13.7)	18.10 ± 6.15		21.61 ± 7.04		36.67 ± 10.38		55.90 ± 13.88	
Work from home									
No	733 (22.2)	17.37 ± 5.37	0.199	21.26 ± 7.26	0.067	34.93 ± 9.38	<0.001	56.26 ± 13.29	<0.001
Yes	2566 (78.8)	17.68 ± 5.87		20.72 ± 7.02		36.65 ± 9.59		59.79 ± 13.43	
Quarantine status									
Never	1440 (43.6)	16.61 ± 5.78	<0.001	20.34 ± 7.50	0.002	34.76 ± 9.65	<0.001	59.19 ± 13.47	0.720
Was quarantined	1329 (40.3)	18.18 ± 5.37		21.28 ± 6.53		36.74 ± 9.27		58.96 ± 13.01	
During quarantine	530 (16.1)	18.91 ± 6.19		21.09 ± 7.11		39.20 ± 9.29		58.65 ± 14.60	

Non-single: married/domestic partnership. Single: single/divorced/widowed/separated. Non-Buddhist: Non-religious/Christian/Muslim/other. Non-Universal Coverage Scheme: state enterprises/social security scheme/self-payment/other. Other occupations: unemployed/retired/farmer/laborer/self-employed/private enterprise/government/state enterprise employee/freelance/other. Not residing with family: residing with others/residing alone. Note: age and quarantine status were subjected to one-way ANOVA. Sex, education level, religion, marital status, occupation status, residence, income, health insurance, history of chronic noncommunicable disease, region of residence and work from home were assessed using *t*-tests.

**Table 3 healthcare-10-00812-t003:** Correlation coefficients and descriptive statistics among variables.

VARIABLE	1	2	3	4	5	6	7
1. Fear of COVD-19	1						
2. Perceived stress	0.176 ***	1					
3. Neuroticism	0.050 ***	0.685 ***	1				
4. Perceived social support-Total	−0.037 **	−0.381 ***	−0.330 ***	1			
5. Perceived social support from significant others	0.007	−0.259 ***	−0.216 ***	0.820 ***	1		
6. Perceived social support from family members	−0.024	−0.385 ***	−0.329 ***	0.815 ***	0.483 ***	1	
7. Perceived social support from friends	−0.076 ***	−0.285 ***	−0.262 ***	0.807 ***	0.498 ***	0.500 ***	1

Note: ** *p* < 0.01.; *** *p* < 0.001.

**Table 4 healthcare-10-00812-t004:** Direct, indirect and total effects on fear of COVID-19 concerning perceived stress in four mediation models (*n* = 3299).

Path	Coeff.	95% LL-CI	95% UL-CI	SE	*p*-Value
**Model 1**					
Total effect (c)	0.123	0.095	0.152	0.014	<0.01
Direct effect (c’)	0.103	0.083	0.123	0.010	<0.01
Fear of COVID—Neuroticism (a1)	0.048	0.01	0.095	0.024	0.047
Fear of COVID—Total perceived social support (a2)	−0.053	−0.123	0.017	0.036	0.140
Neuroticism—Perceived stress (b1)	0.347	0.330	0.363	0.08	<0.01
Total perceived social support—Perceived stress (b2)	−0.074	−0.085	−0.063	0.05	<0.01
Neuroticism—Total perceived social support	−0.349	−0.382	−0.316	0.017	<0.01
Total indirect effect	0.020	0.02	0.039	0.010	0.032
Fear of COVID—Neuroticism—Perceived stress	0.018	0.00	0.036	0.09	0.047
Fear of COVID—Perceived social support—Perceived stress	0.08	−0.03	0.019	0.06	0.141
**Model 2**					
Total effect (c)	0.123	0.095	0.152	0.014	<0.01
Direct effect (c’)	0.107	0.086	0.127	0.010	<0.01
Fear of COVID—Neuroticism (a1)	0.048	0.01	0.095	0.024	0.047
Fear of COVID—Perceived social support from significant others (a2)	0.08	−0.021	0.036	0.015	0.60
Neuroticism—Perceived stress (b1)	0.370	0.353	0.386	0.08	<0.01
Perceived social support from significant others—Perceived stress (b2)	−0.117	−0.142	−0.092	0.013	<0.01
Neuroticism—Perceived social support from significant others	−0.214	−0.246	−0.181	0.017	<0.01
Total indirect effect	0.017	−0.02	0.035	0.09	0.074
Fear of COVID—Neuroticism—Perceived stress	0.018	0.00	0.036	0.09	0.047
Fear of COVID—Perceived social support from significant others—Perceived stress	−0.02	−0.09	0.05	0.04	0.599
**Model 3**					
Total effect (c)	0.123	0.095	0.152	0.014	<0.01
Direct effect (c’)	0.104	0.084	0.124	0.010	<0.01
Fear of COVID—Neuroticism (a1)	0.048	0.01	0.095	0.024	0.047
Fear of COVID—Perceived social support from family members (a2)	−0.014	−0.043	0.015	0.015	0.348
Neuroticism—Perceived stress (b1)	0.351	0.334	0.368	0.08	<0.01
Perceived social support from family members—Perceived stress (b2)	−0.176	−0.203	−0.150	0.014	<0.01
Neuroticism—Perceived social support from family members	−0.317	−0.350	−0.284	0.017	<0.01
Total indirect effect	0.019	0.01	0.038	0.09	0.043
Fear of COVID—Neuroticism—Perceived stress	0.018	0.00	0.036	0.09	0.047
Fear of COVID—Perceived social support from family members—Perceived stress	0.05	−0.05	0.015	0.05	0.348
**Model 4 ***					
Total effect (c)	0.123	0.095	0.152	0.014	<0.01
Direct effect (c’)	0.10	0.080	0.121	0.010	<0.01
Fear of COVID—Neuroticism (a1)	0.048	0.01	0.095	0.024	0.047
Fear of COVID—Perceived social support from friends (a2)	−0.047	−0.074	−0.019	0.014	0.01
Neuroticism—Perceived stress (b1)	0.361	0.344	0.378	0.08	<0.01
Perceived social support from friends—Perceived stress (b2)	−0.131	−0.160	−0.102	0.015	<0.01
Neuroticism—Perceived social support from friends	−0.326	−0.358	−0.294	0.016	<0.01
Total indirect effect	0.023	0.05	0.042	0.09	0.013
Fear of COVID—Neuroticism—Perceived stress	0.018	0.00	0.036	0.09	0.047
Fear of COVID—Perceived social support from friends—Perceived stress	0.016	0.06	0.025	0.05	0.01

Controlling covariates: age, sex, education level, religion, occupation status, history of chronic non-communicable disease and quarantine status. Note: * the parallel mediation model was significant (*p* < 0.05). Coeff. = unstandardized coefficients, LL-CI = lower limit confidence interval, UL-CI = upper limit confidence interval, SE = standard error.

## Data Availability

Upon reasonable request, the author will contact the director of the HOME-COVID-19 survey study and provide the data, given his or her consent.

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
