# Peer review of "Fear of COVID-19 and Perceived Stress: The Mediating Roles of Neuroticism and Perceived Social Support"

_healthcare, 2022, doi:10.3390/healthcare10050812_

Round 1

Reviewer 1 Report

This is a very interesting article presenting the results of research conducted during the Covid-19 pandemic (Fear of COVID-19 on Perceived Stress: Mediator Roles of Neuroticism and Perceived Social Support). The design of the article corresponds to the texts that are based on empirical studies and contains all the elements of the structure of this type of study. The terminology is explained comprehensively based on the properly selected literature on the research subject. The research objective has been achieved and the presented results are cognitively interesting and prove good methodological skills of the authors. The strength of the study is the large sample and meticulousness of the quantitative analyses presented.
Suggestions for authors:
1. I propose to separate the purpose and hypotheses of the research. 
2. Suggests a separate section:  practical implications.

Author Response

Psychotherapy Unit & Geriatric Psychiatry Unit,

Department of Psychiatry, 

Faculty of Medicine, Chiang Mai University,

Chiang Mai,

Kingdom of Thailand. 50200

Dear Editor,

Re: Reviewers’ comments on manuscript ID healthcare-1689209, Fear of COVID-19 on Perceived Stress: The Mediating Role of Neuroticism and Perceived Social Support, dated 15th Apr 2022

20th Apr 2022

Dear Editor,

Thank you for your useful comments and suggestions. We have revised the manuscript as suggested.  Please find the summary of the revisions point-by-point presented below.

Response to the comment from Review#1:

  1. I propose to separate the purpose and hypotheses of the research.

Response: We have separated the purpose  and hypotheses of the research as suggested by adding the texts We hypothesized that there were both direct and indirect effects of fear of COVID-19 concerning perceived stress through the two mediators in the general Thai population.’ in the last paragraph of Introduction section. Please see page 3.

  1. Suggests a separate section:  practical implications.

Response: Thank you for pointing this out. We had added a paragraph stated practical implications as follow:

Practical implications

In order to reduce perceived stress as a result of fear of COVID-19 during the beginning of the pandemic, governments must provide sufficient and accurate information about the transmission and preventive actions to the public to counterbalance false information that triggers more fear. In addition, all sectors of society should carry out education campaigns to further disseminate public health information and prevent the spread of the disease, particularly, signs of perceived stress should be educated to the public. If this was done, people could easily and early identify the perceived stress experienced by themselves and their family, friends, and social network. In addition, healthcare providers, family, and friends should observe and monitor stress from fear of COVID-19, particularly in individuals at high risk of stress, such as individuals with neuroticism personality traits, and prioritize support to these individuals. Although, there was some limitation of social gathering during the pandemic, for young adults, it is important to keep connections and communication with close friends and family members through available methods (e.g., social media) to provide mutual support. Political leaders should organize alternative quarantine protocols for people who experienced fear and stress, and arrange for them to stay with family or friends. Moreover, cooperation between family, friends, and significant others, including schoolteachers, counselors, colleagues, healthcare providers, and community leaders to provide instrumental, emotional support, and reliable information support to reduce stress during the pandemic are crucial to prevent future mental health problems.’ Please see page 17.

Response to the comment from Review#2:

  1. Title: Fear of covid....: the mediating role of neuroticism...

Response: Thank you for your suggestion. We have revised the title to ‘Fear of COVID-19 on Perceived Stress: The Mediating Role of Neuroticism and Perceived Social Support’ as suggested.

  1. Line 31: there is an underlined sentence. It is probably an error.

Response: Thank you for pointing this out.  Underline has been deleted.

  1. Line 64: I suggest underlining that stress does not have a negative effect only on those who are predisposed to psychological health problems but also on the general population with effects for example on sleep (https://doi.org/10.1016/j.eclinm.2021.100916), emotional eating (https://doi.org/10.3390/jpm11060569) and weight gain (https://doi.org/10.1111/cob.12453)

Response: Thank you for your advice. We have added the following: Stress affects those who are predisposed to psychological health problems, but also the general population (e.g., it can affect sleep, emotional eating, and weight gain). and references as suggested. Please see page 2, the 2nd paragraph.

  1. Line 98: True, no studies have looked at perceived social support as a mediator of the fear-stress relationship. However, there is some evidence that social support plays a mediating role in the relationship between fear and mental health outcomes (https://doi.org/10.1007/s12144-021-02395-y). I think it's useful to keep this reference in mind for discussion.

Response: Thank you for your advice and the citation. It’s an interesting article, indeed. Unfortunately, we came across serious flaws in it. We read them carefully and enthusiastically; however, we queried that the hypothesized model and the findings were conflicting. We found several spots that might need further investigations. First, according to their model using AMOS, Fear of COVID-19 has a role as a mediator, and Support did not.  

Mahamid et al, 2021.

As shown by the figure (above) (Mahamid et al, 2021.), this is the mediation model where “COVID Fear” serves as a mediator rather than social support as the researchers intended. Second, there is a phrase, ‘Concerning the moderating effects (H1),…’ These two pieces of information confused us because they intended to test the mediating effects (not moderating effects). Third, we were confused by these sentences: ‘However, this effect was made up of a statistically significant indirect effect (via social support, βX, M, Y = -. 286; p < 0.01). Normally, the indirect effect is not shown on the path model - unlike the direct effect. Calculating by multiplying the direct effect of a path and the direct effect of the b path would give the results of the indirect effects. Finally, the conclusion that ‘the relationship between fear of COVID-19 and psychological distress was fully mediated by social support’ was not justified by the data.

We apologize that we cannot include this citation as suggested because of the conflicting results that need to be amended. We hope you agree.

  1. Line 99-103: did the authors have a predetermined hypothesis? if so, please specify.

Response: Yes, we did. Hypotheses have been added. Please see the texts,

Hypothesized model

We hypothesized that there were positive correlations between fear of COVID-19, perceived stress, and neuroticism, while there were negative correlations between fear of COVID-19 and perceived social support, between perceived social support and perceived stress, and between perceived social support and neuroticism. We also hypothesized that there were a significant direct effect of fear of COVID-19 and perceived stress, and a significant indirect effect of fear of COVID-19 and perceived stress through neuroticism and perceived social support (parallel mediation). (Figure 1)’ in Data analysis subsection, page 5.

  1. Methods: was there a maximum age limit? How were participants recruited? Through email, social network ads, word of mouth?

Response: Regarding the HOME-COVID-19 project (Nochaiwong et al., 2020), there was no maximum age limit; however, our analysis focused only on adult population aged between 18-59 years old. We have rewritten the first and second paragraph of Materials and Methods (page 3) as follows:  

‘The present study employed a secondary data analysis of The Health Outcomes and Mental Health Care Evaluation Survey: Under the Pandemic Situation of COVID-19 (HOME-COVID-19) which was a cross-sectional survey with Wave I data collected from April 21st to May 4th, 2020, among 4,004 participants comprising a general population in Thailand [7] . Participants aged 18-59 years old were included in this analysis.

The HOME-COVID-19 study

This was an open, voluntary, and nationwide cross-sectional online survey using the SurveyMonkey® platform in Thailand. Participants included Thai citizens, permanent residents and nonresidents with employment or work permits, aged 18 years or above at the time of the survey. The study recruited participants using convenience sampling and a snowball technique by posting the survey QR codes or links to social media networks and public websites, such as Facebook, , Twitter and Instagram. The study was in line with the Strengthening the Reporting of Observational Studies in Epidemiology Statement [54] and Improving the Quality of Web Surveys: The Checklist for Reporting Results of Internet E‐Surveys [55]. Details of the methods of HOME-COVID-19 have been published elsewhere [7].’

  1. Line 117: there is an underlined dot.

Response: Thank you for your careful review. The underlined dot has been deleted.

  1. Line 123: please specify if this is a validated measure and who developed it.

Response: Thank you for the queries. Please be informed that Fear of COVID-19 and its impact on Quality of Life is new and was developed for its use after COVID-19 had emerged in Thailand for about 2 months. The developers of this tool were me (TW) and the other co-author (NW-the 7th author of the current study). Both of us were researchers in the HOME COVID-19 study. Although we have been doing research in developing measures in mental health, psychiatry, and behavioral science for about 20 years, due to the limited time before its use, the psychometric properties of the tool were not completely assessed. We published its internal consistency from a pilot among 30 participants and reported them as .925 and .911 in the fear part and the quality of life part, respectively. More complete information can be found at http://www.wongpakaran.com/index.php?lay=show&ac=article&Id=2147599077 . Therefore, we have this section rewritten as follows:

‘The scale comprises two parts, namely, fear of COVID-19 (9 items) and impacts on quality of life (QoL) (8 items). Regarding the fear of COVID-19 (FOC) part, each item on a five-point Likert type of scale, ranging from 4 (I am the most fearful) to 0 (I'm not fearful at all), indicated the level of fear for each condition. A higher score indicates a higher FOC. For the Impact on QoL part, each item on a five-point Likert type of scale, ranging from 4 (Most impacted) to 0 (Least or not at all impacted), indicated the level of impact of the fear. A higher score indicates a higher impact of fear. The total scores range from 0 to 68 with 0-36 in the FOC part and 0-32 in the impacts on the QoL part. A pilot study among 30 adult Thai participants demonstrated Cronbach’s alphas of .925 and .911 in the FOC parts and impacts on QoL, respectively [56]. In the present study, Cronbach’s alpha of FOC was .878.’. Please see Materials and Methods, Instrument subsection, 2.1.1, page 3.

  1. Line 136: specify if this is a validated measure.

Response: Thank you. We have this part rewritten. Please see the texts, The 10-item scale was developed to measure perceived stress.[57] It contains two factors -- i.e., 6 items for stress and 4 items for control. Each item is scored on a five-point Likert type of scale, ranging from 4 (very often) to 0 (never), indicated the frequency of feeling or thinking a particular way during the past month. A higher score indicates higher perceived stress, and scores range from 0 to 40. The Thai version was validated with Cronbach’s alpha of .85, excellent goodness-of-fit and good validity [21]. Cronbach’s alpha for the present study was .821.’, page 3.

  1. Are the reported Cronbach alphas from the original validation study or from the current study? Please specify.

Response: We clarified the Cronbach’s alphas in each instrument. Please checked with the revised information.

  1. Line 154: here the authors mentioned that participants with age >60 were excluded, please specify this aspect also in the method section.

Response: Thank you. We added the texts, Participants aged 18-59 years old were included in this analysis.’ In the Materials and Methods. Please see it in the first paragraph, page 3.

  1. Several factors not mentioned in the method section were included in the table that displayed sociodemographic characteristics. Please note that in addition to the questionnaires, a self-report form (?) was used to assess marital status, educational level, etc. In addition, were participants required to state which non-communicable disease they had?

Response: Thank you for pointing this out. A paragraph, Sociodemographic data were collected by self-report -- i.e., age, sex, marital status, education level, income, religion, regions of residence, occupation and work status, health status, as well as living status (e.g., being in quarantine).’, was added to the Measurements part. Please see page 3.

  1. In the discussion section a lot of ‘’related research’’ was used, please use also other expressions like ‘’previous evidence’’, ‘’a study by…’’ ‘’findings from other studies…’’, and so on.

Response:  Thank you for your suggestion. We have revised those terms throughout the Discussion section.

  1. Line 365: the sentence ‘’ this tended…’’ is not clear.

Response: We have revised the sentence to Neuroticism was also found to be the most significant factor influencing the perception of threat from Coronavirus. Please see Discussion section, page 15, last paragraph.

  1. Finally I think it would be interesting to broaden the discussion a bit to speculate on what the reasons might be that only social support of friends is significant.

Response: Thank you for your thoughtful suggestion. We have added an additional discussion section speculating the reasons that only social support of friends was found significant in the present study as ‘Family and friends provide practical and emotional support when individuals need help coping with stressful life events [96]. Particularly during the COVID-19 pandemic, the government had implemented social distancing policies (e.g., work from home, online study, or even lockdown) [28,75,97]. Subsequently, people spent more time with their family at home or with friends who live together. For young adults, social support from a friend has an important impact on the psychological effects of the pandemic. Previous studies with Chinese students found that peer support was an important source of social support for college students [82,83]. We speculate that perceived social support from family did not moderate the relationship between fear of COVID-19 and perceived stress, but rather had a direct effect on perceived stress due to high perceived social support in Thai adults in the present study even during the pandemic. A study with Thai adults during the early phase of the COVID-19 pandemic suggested that about 40-58% of the participants reported better relationships with family by providing care, communicating, offering emotional support, supporting problem-solving for each other, and participating in family activities [98]. Direct association of perceived social support from family on perceived stress during the pandemic in adult populations has been reported in adult populations in different countries [79,99]. In addition,’. Please see page 16.

Hopefully, our revision will be sufficient and satisfy the editor and reviewers. Additional corrections were made by deleting the duplicate sentences in the Discussion section. We have corrected many other minor errors and misspellings throughout the manuscript.

Thank you for your consideration again. We are looking forward to hearing from you soon.

Best regards,

Prof. Tinakon Wongpakaran, MD, FRCPsychT

Reviewer 2 Report

This is a good study, well-conducted, with interesting results that help us better understand the factors that underpin the relationship between fear of covid and stress. The introduction is well-written, concise, and easy to understand. Some things were not specified in the methods section, which need to be improved. The discussion section is well-written.

I suggest the following revision:

Title: Fear of covid....: the mediating role of neuroticism... 

Line 31: there is an underlined sentence. It is probably an error.

Line 64: I suggest underlining that stress does not have a negative effect only on those who are predisposed to psychological health problems but also on the general population with effects for example on sleep (https://doi.org/10.1016/j.eclinm.2021.100916), emotional eating (https://doi.org/10.3390/jpm11060569) and weight gain (https://doi.org/10.1111/cob.12453)

Line 98: True, no studies have looked at perceived social support as a mediator of the fear-stress relationship. However, there is some evidence that social support plays a mediating role in the relationship between fear and mental health outcomes (https://doi.org/10.1007/s12144-021-02395-y). I think it's useful to keep this reference in mind for discussion.

Line 99-103: did the authors have a predetermined hypothesis? if so, please specify.

Methods: was there a maximum age limit? How were participants recruited? Through email, social network ads, word of mouth?

Line 117: there is an underlined dot.

Line 123: please specify if this is a validated measure and who developed it.

Line 136: specify if this is a validated measures.

Are the reported Cronbach alphas from the original validation study or from the current study? Please specify.

Line 154: here the authors mentioned that participants with age >60 were excluded, please specify this aspect also in the method section.

Several factors not mentioned in the method section were included in the table that displayed sociodemographic characteristics. Please note that in addition to the questionnaires, a self-report form (?) was used to assess marital status, educational level, etc. In addition, were participants required to state which non-communicable disease they had?

In the discussion section a lot of ‘’related research’’ was used, please use also other expressions like ‘’previous evidence’’, ‘’a study by…’’  ‘’findings from other studies…’’, and so on.

Line 365: the sentence ‘’ this tended…’’ is not clear.

Finally I think it would be interesting to broaden the discussion a bit to speculate on what the reasons might be that only social support of friends is significant.

Author Response

(The authors gave the same response as above.)
